# Mapping the Literature on Nutritional Interventions in Cognitive Health: A Data-Driven Approach

**DOI:** 10.3390/nu11010038

**Published:** 2018-12-24

**Authors:** Erin I. Walsh, Nicolas Cherbuin

**Affiliations:** Centre for Research on Ageing, Health and Wellbeing, Research School of Population Health, Australian National University, Canberra 0200, Australia; nicolas.cherbuin@anu.edu.au

**Keywords:** citation network analysis, text mining, nutrition intervention, cognition

## Abstract

Manual review of the extensive literature covering nutrition-based lifestyle interventions to promote healthy cognitive ageing has proved educational, however, data-driven techniques can better account for the large size of the literature (tens of thousands of potentially relevant publications to date) and interdisciplinary nature, where relevant publications may be found. In this study, we present a new way to map the literature landscape, focusing on nutrition-based lifestyle interventions to promote healthy cognitive ageing. We applied a combination of citation network analysis and text mining to map out the existing literature on nutritional interventions and cognitive health. Results indicated five overarching clusters of publications, which could be further deconstructed into a total of 35 clusters. These could be broadly distinguished by the focus on lifespan stages (e.g., infancy versus older age), and specificity regarding nutrition (e.g., a narrow focus on iodine deficiency versus a broad focus on weight gain). Rather than concentrating into a single cluster, interventions were present throughout the majority of the research. We conclude that a data-driven map of the nutritional intervention literature can benefit the design of future interventions, by highlighting topics and themes that could be synthesized across currently disconnected clusters of publications.

## 1. Introduction.

The literature surrounding nutrition interventions intended to prevent cognitive decline in ageing is large, multi-faceted and heterogeneous. This reflects the heterogeneity of ageing, and the large variety of methods, participants, and intervention targets of nutrition interventions to date. Systematic reviews and meta-analyses are key tools in addressing this heterogeneity and synthesizing large numbers of publications into useful formats. Systematic reviews carried out by human researchers are necessarily targeted at specific research questions, and so tend to be highly specialized. Broadly, they focus on literature with a scope intentionally limited to a topic that a human reader could explore and meaningfully summarize. Examples of this are reviews focusing on intervention delivery (e.g., computer-tailored promotion of adherence [1]), specific nutritional content (e.g., interventions focused on flavonoids [2], antioxidants [3], polyphenols [4], or dietary patterns such as the Mediterranean diet [5]), particular demographics (e.g., children and adolescents [6], university students [7], older adults [8] or Australian adults [9]) or clinical populations (e.g., cancer patients [10]), and specific cognitive outcomes (e.g., the development of Alzheimer’s disease [11] or dementia [12]).

From strict meta-analyses to a more narrative approach, these focused manual reviews are extremely valuable in organizing and conveying knowledge relating to any of these particular domains. Yet, they are inherently limited by any given author or team’s understanding of the existing literature, which leads to the selection of topics on the basis of an incomplete picture of relevant knowledge [13]. This results in a constellation of specialized knowledge without clear synthesis or links between the topics, which does little to reveal new or underserved topics, which fall beyond the *a priori* scope of existing reviews. This is a reflection of the sheer breadth of available information. Even with increasingly sophisticated search and aggregation tools, it is impossible for an individual or team to manually identify and synthesize every relevant paper from peer-reviewed literature, which expands by over one million publications per year [14,15].

Accordingly, increasing attention is being paid to automated methods for knowledge identification, synthesis, and summary [16]. Recent advances in citation network analysis and text mining software provide new opportunities for constructing robust summaries of the literature and concepts therein, by a purely data-driven approach [17,18]. Clusters formed by groups of publications connected by mutual citation can be taken as indicative of theoretical or conceptual groupings in the literature [17]. Text-mining techniques, including topic models and semantic neighborhood analyses, are increasingly being used to extract meaning from lengthy passages of text [19]. 

In this paper, we present a new way to map the literature landscape focusing on nutrition-based lifestyle interventions to promote healthy cognitive ageing. We introduce a data-driven approach that: draws from citation networks, abstract and title text; allows efficient synthesis of more publications than a human could manually identify and read; and demonstrate how this approach can be applied to give new insights into the themes and gaps present in the intervention literature. We conclude that a data-driven map of the nutritional intervention literature can benefit the design of future interventions, by highlighting topics and themes that could be synthesized across currently disconnected clusters of publications.

## 2. Materials and Methods 

The citation network analysis applied here aims to map scholarly literature and identify citation clusters relevant to a specific topic (e.g., nutritional interventions to improve cognitive health). It involves: (1) systematically searching the literature for pre-defined search terms and obtaining mutual citation links, full title and abstract text for this literature, (2) conducting citation network analysis on mutual citation links to identify clusters within this literature, and (3) using text mining techniques to characterize these clusters.

### 2.1. Literature Search and Citation Network Analysis

CiteNetExplorer [20] software is a tool for visualizing and analyzing citation networks denoted by mutual citation, including relatedness between papers as a weighted combination of the year of publication and mutual citation, and identification of ‘clusters’ of publications. These are logical groupings of publications located near to one another in the larger citation network, as established by a variant of the modularity function (described in [21]). A Web of Science Core Collection Database search was undertaken on 23 October 2018 for the terms *(((cognit* OR dementia) AND (ageing OR aging) NOT (animal)) AND (diet OR nutr*))*, with a restriction to peer-reviewed journals. Note the absence of ‘nutrition’ and ‘diet’, as the intention is to later examine the position of these terms within the revealed clusters. This yielded 6138 citations. This was reduced to 6045 once the search was restricted to peer-reviewed journal texts. Full records of citations and secondary articles (those citing and cited by the documents) were imported into CiteNetExplorer for cluster analysis. Cluster analysis was undertaken on the remaining 6045 publications (minimum cluster size set to 10 publications and 10 iterations from the random seed 1337). To obtain detail, this process was repeated iteratively until larger clusters (*n* > 500 publications) could not be further deconstructed into smaller clusters. Some citations were omitted in this process as not clearly belonging to any particular cluster, or due to missing information, resulting in a final *n* = 4915 (see Appendix A).

### 2.2. Text Preparation

All titles and abstracts were extracted from Web of Science search results, which provide digital object identifiers (doi) and additional details. When unavailable from the search, titles and abstracts were reconstructed from doi, author and year, via a composite of automated python script: Elsevier scopus Application Programming Interface (APC) via ‘fulltext’ package in R (version 1.01) and ‘roadoi’ package in R (version 0.5.2; *n* = 2372 titles and abstracts) and manual entry (*n* = 221 titles and abstracts) referring to doi.org and Google Scholar. The resultant titles and abstracts were converted into a corpora (collection of natural language documents) in the tm package (version 0.6-2 [22]). Following text mining convention [23], all non-word information (stop words, case, punctuation, case, etc.) was removed. Words were stemmed using Porter’s stemming algorithm (e.g., “cognition”, “cognitive” become “cognit”; “diet”, “dietary” become “diet”). The resultant corpora was saved as spreadsheets and as term-document matrices (TDM), which described the frequency of terms (columns) occurring across the clusters (rows).

### 2.3. Cluster Description

The simplest method of exploring a cluster is the generation of word clouds. These provide a parsimonious visual overview of terms found within a text, with size and opacity indicating the frequency of a word within the text. Building on this, latent Dirichlet allocation (LDA) is a Bayesian topic model which probabilistically extracts topics from terms across documents (here, manuscript titles). LDA treats each term as a finite mixture of possible underlying topics. This is expressed as beta (β), the probability of that term being generated by the topic. The log ratio of β for one topic as opposed to another, obtained by log(β_topic1_/β_topic2_) can be used to identify terms most distinctive to each topic, which therefore describe it best. Topic modelling was carried out on the title of each cluster using the topicmodels package for R (version 0.2-6, [24]).

There are multiple techniques available should the reader wish to go further and establish the context of a particular term (such as “intervention”). Here, we provide an example of how pairwise associations in text (‘findAssocs’ function of tm) and neighbourhood analysis (‘neighbors’ function of the LSAfun package (version 0.5.1 [9])) of abstract text, within a cluster, can provide this context. Using these techniques in combination, capture both physical proximity of words in the text (e.g., “Happy” next to “Child”, “Joyful” next to “Adult”), and semantic proximity (e.g., “Happy” will be closer to “Joyful”, while “Child” is closer to “Adult”). 

## 3. Results

Word clouds (Figure 1) provide a visual overview of the 6045 journal articles published from 1929 to 2018, relating to cognition, ageing, nutrition/diet, and interventions, grouped by the cluster. In aggregate, recurrent terms throughout denote publications that tend to focus on lifespan stages (with a particular focus on childhood and older age), with studies focusing on ‘patients’ more common in older age. While some clusters reflect a very specific focus (e.g., publications surrounding phenylketonuria or iodine levels are quite distinct), there is clear overlap across the wider literature with terms such as ‘develop’, ‘outcome’ and ‘function’ being present throughout. Cluster descriptions revealed by topic analysis (Figure 2; see Appendix A for more detail) demonstrates how text mining approaches can more clearly map out a large, diffuse literature such as this. 

Topic analysis of the five main clusters and 30 sub-clusters reveal several insights into the landscape of the literature addressing lifestyle dietary intervention to improve cognition (Figure 2; Appendix A). Broadly, clusters form around lifespan stages: Two clusters focus on older age (cluster 1: The association between diet and cognitive outcomes with a focus on the prediction of decline and disease; cluster 2: Daily self-care and nutrition in older age), two on lifespan or midlife (cluster 4: The role of diet in overweight and obesity throughout the lifespan; cluster 5: Diet and phenylketonuria), and one in childhood (cluster 3: The association between nutrition in early life and subsequent cognition). There are clear sub-literatures within larger topics, e.g., *maternal diet and breastfeeding outcomes* (3ba) within *breastfeeding and cognitive outcomes* (3b) and within *the association between nutrition and early life and subsequent cognition* (cluster 3). Yet, the tendency for publications to cluster around the time of life precludes what might be considered logical sub-cluster groupings. For example, due to conceptual similarities, one might expect sub-clusters relating to antioxidants, choline, trace metals such as magnesium and iodine to fall within close proximity. However, they are more specific to the adult-specific cluster 1 (antioxidants 1c; choline 1d) and child-specific cluster 3 (magnesium 3aab; iodine 3d).

The term ‘intervention’ is present in the abstracts of all but 3 of the total 35 possible clusters. Text mining (Appendix A) reveals that interventions are largely cluster-specific in terms of methods and focus. For example, in sub-cluster *breastfeeding and cognitive outcomes*, ‘intervention’ correlates with ‘baby friendly’ and ‘characteristic adjustment’, and is semantically near terms such as ‘computerized’, ‘instrumental’ and ‘intention’, reflecting interventions in this cluster target breastfeeding indirectly via intention (as it would be unethical to randomly assign a child who otherwise would have been breastfed to a formula condition). Conversely, in sub-cluster *dietary restraint and weight loss* (4b) ‘intervention’ correlates with terms ‘prevent’, ‘trial’, and ‘random’, and is semantically near terms such as ‘weight’, ‘pertaining’, ‘targeted’, reflecting interventions in this cluster directly target behaviors relating to weight gain.

## 4. Discussion

Nutrition interventions to prevent cognitive decline in ageing are extremely varied in terms of sample, approach, and focus. They are also highly numerous. Even if knowledge is collected and curated in the form of meta-analyses or discursive reviews, the sheer size of the literature makes it impossible for a single researcher or team to manually construct an overview of extant trends, syntheses, and gaps in knowledge. In this study, we demonstrated a method for efficient synthesis of a large number of publications and produced a map of the literature regarding nutrition-based lifestyle interventions to promote healthy cognitive ageing. This map can be used to characterize the intervention literature as a whole, identify thematic overlap between work that has to date remained separate, and identify gaps requiring further study.

This approach can also be used to relatively quickly select a large number of studies, which address a common topic. Based on the cluster identified in Figure 2, it is possible to retrieve all the related articles for more detailed review (Appendix A). This functionality can be used in its own right to survey a particular part of the literature or to supplement searches implemented in systematic reviews or meta-analyses. These approaches are complementary because the purpose of reviews and analyses are to distill concepts, which offers precision but may exclude relevant targets, while the purpose of citation network analysis is to connect concepts and uncover a broader scope of relevant targets. In particular, the inclusion of second-degree connections (citations of papers included in search responses, which themselves may have not been included in the original search) can uncover linkages that manual review overlooks.

Beyond accessing knowledge in clusters, the mapping of the extent of literature can be used to identify hierarchical features that provide insights on how content of interest can be better identified. In the current context, we found that the nutrition intervention literature first formed clusters on the basis of participant age (notably childhood and late life), before further subdivision into nutritional content of the intervention. In some instances, this makes intuitive sense; prenatal nutrition (cluster 4d) is unique to very early life, while frailty (cluster 2aa) is an issue inherent in old age. However, the association between nutrition is often lifelong, and conceptual segregation based on age is not necessarily helpful. This can be seen in the topics of the clusters, for example, the theme of obesity is split into youth (*The role of diet in overweight and obesity in children and adolescents*, cluster 4) and old age (*Psychopathology in adults with obesity*, cluster 2d), and in individual publications. For example, Hamadani et al., [25] and the Supplémentation en Vitamines et Minéraux Antioxydants, Su.Vi.Max study [26] differ in terms of size (*n* = 168 vs. *n* = 13,017, respectively), location (Bangladesh vs. France), participant age and measures of cognition (3–13 month infants and the Bayley scales of infant development in vs. 45–60 year-old adults and the RI-48 cued recall test, respectively). They do not share a single citation in common, indicating that they come from disconnected portions of the literature [17]. Accordingly, they are situated in separate clusters (*Trace Metals and Cognition in Children,* aab, and *Alzheimer’s Disease in Ageing populations,* 1aaa, respectively). For the purposes of parsimony, the Su.Vi.Max project description is cited here; to the author’s knowledge at the time of writing this statement is true for all subsequent publication of Su.Vi.Max data.

Yet, both studies include the examination of the impact of zinc supplementation on cognitive outcomes, which relies on biological processes that are relevant across the lifespan [27]. Even accounting for the possibility that there is little equivalence between these interventions beyond the inclusion of zinc supplementation, their conceptual isolation is striking.

This clustering on the basis of age, rather than conceptual content, is not a shortcoming of our analysis, but rather an insight into the genuine structure of the nutrition interventions as they currently sit within the literature. A single citation can sit within multiple clusters, therefore it is unlikely that the cluster structure we found is because single interventions include multiple targets (e.g., the Su.Vi.Max study [26] includes both antioxidant and trace metal supplementation). Instead, it is more likely that this structure arises because of the relatively small number of interventions explicitly targeting a lifespan perspective [28], while the majority of studies select specific age ranges to allow clear measurement and interpretation of cognitive outcomes. This could implicitly lead to researchers consulting only age-relevant literature, particularly if their searches focus on interventions because cross-sectional observational studies often encompass wider age ranges. The situation could be likened to the citation siloes formed on the basis of language barriers in scholarly literature, prior to widespread translation services [29]. Fortuitously, the solution to avoiding redundancy and improving knowledge synthesis, in this case, is simpler and more immediate. Once it is clear that the literature is organized in this way, future interventions where a nutritional factor is relevant across the lifespan can be improved by conducting searches, which mindfully incorporate the whole lifespan.

The current approach can also be applied to identify gaps in the literature. It is beyond the scope of this manuscript to exhaustively identify gaps in the literature as revealed by the map, but there are some trends of note. Given the age-based structure of the literature, the most evident gap is the lack of specific clustering of interventions around mid-life, when many risk factors develop and start having a measurable impact on brain and cognitive health [30,31]. Other potential gaps may include interactive effects with genetics, social and environmental variables as well as contributions of ethnic, cultural and socio-demographic factors. There are also some gaps, which are more likely to be indicative of the search terms used. This emphasizes that the choice of search terms is as important in this method as they are for reviews or meta-analyses. Interested readers will note omissions of clusters on known correlates of nutrition and cognitive outcomes, such as type 2 diabetes [32], and no clusters forming around terms such as ‘neuroimaging’ or ‘brain’. This may be due to the choice of search terms insensitive to these topics, or possibly the reliance on titles, rather than full text, for topic analysis. The latter is likely because the terms ‘diabetes’, ‘neuroimaging’, and ‘brain’ were detected as correlates of ‘nutrition’ when text mining was undertaken on abstracts, which by virtue of their length and purpose convey more context. This could be further investigated by conducting topic analysis on the full texts of manuscripts, though this process may prove challenging to automate due to barriers of copyright for full-text access.

The data-driven approach we present here has a number of strengths. Chiefly, we have demonstrated knowledge synthesis and mapping on a scale that reflects the size of the literature, rather than what is possible for a human to achieve. The map produced provides an intuitive overview of the literature and organizes the long list of publications into a format conducive to further reading. Notably, the map we produced is open to refinement. New, more specific searches could be conducted to map a smaller part of the literature with greater precision. However, our findings also highlight genuine avenues for the improvement of nutrition and cognition intervention literature.

## 5. Conclusions

We have presented a novel, data-driven combination of citation network analysis and text mining in order to map the current literature surrounding nutritional interventions on cognitive health. We showed a tendency for mutual citation clusters on the basis of age group (in particular, separation of work pertaining to children and the elderly), before topic. We also noted that interventions are an integral part (rather than separate cluster) of the wider literature. We suggest that future interventions could benefit from researchers reading beyond their target age groups and possibly benefit from topic-relevant insights obtained from interventions carried out at other times in life.

## Figures and Tables

**Figure 1 nutrients-11-00038-f001:**
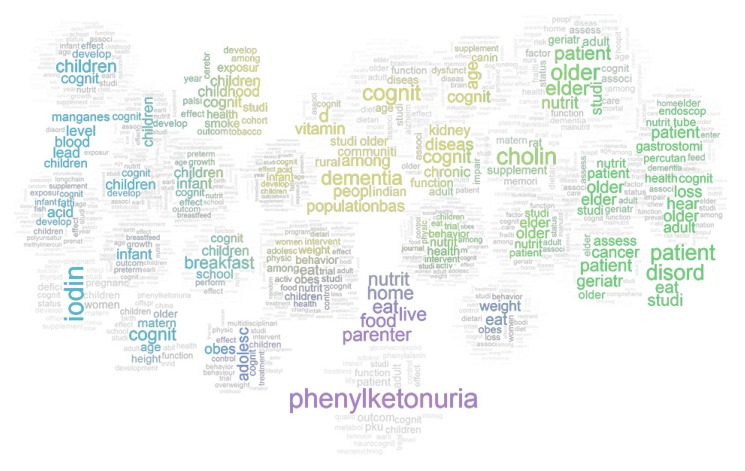
Visual overview of 6045 publications. Note. Word clouds are derived from the frequency of terms within titles, grouped by clusters (described in more detail in Figure 2 and Appendix A).

**Figure 2 nutrients-11-00038-f002:**
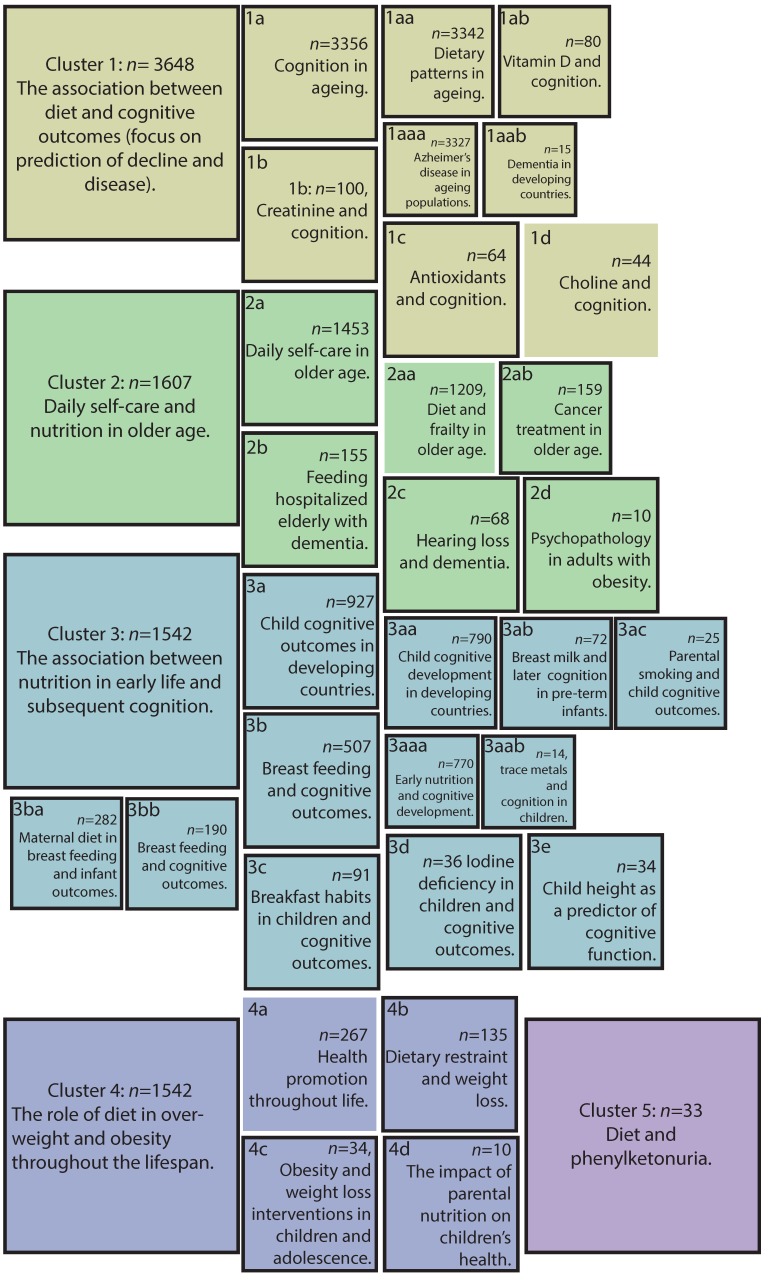
Overview of topic analysis results. *Note.* Each panel depicts a cluster as described by topic analysis undertaken on *n* manuscript title text (see Appendix A for more detail). Alphabetical designations (1a, 1aa etc) have been assigned to clarify the relationship between clusters when referred to elsewhere in text and tables. Size denotes nesting. Black borders indicate that pairwise word correlations with the term “intervent*” are present at *r* < 0.5 and are present with terms in that cluster’s abstracts. See Appendix A for further details on the topic analysis process, and characterization of each cluster and sub-cluster.

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
