# Peer review of "Mapping the Literature on Nutritional Interventions in Cognitive Health: A Data-Driven Approach"

_nutrients, 2018, doi:10.3390/nu11010038_

Reviewer 1 Report

This manuscript presents a systematic review of the nutritional interventions in cognitive health. The manuscript is generally well written, but it is unclear what this review adds to what is already known and have been published earlier. No clear research question seems to be formulated, the conclusions are unclear and other major concerns with this manuscript. 

My specific comments are stated below. Overall, several important issues need to be addressed and some are of methodological character which requires a considerable revision of the paper. 

1. I recommend formulating a specific research question and performing a meta-analysis to answer the question.

2. Methods: Was the study registrered at PROSPERO? 

3. Methods - literature search and selection: Please outline the exact search string or provide an appendix with the search strategy with specific search outcomes for each search and combinations. 

4. Methods - literature search and selection: Did you restrict study selection on any language? 

5. Methods. The authors have not performed a systematic review, according to international standards PRISMA Guidelines, so they do not provide specific numerical data.

6. Results: Please, provide clear results and describe them. Use appropriate statistics. 

7. Discussion, outline your results, discuss their novelty and their application to practice.

8. Conclusions need to be softened, modified a in order to reflect only the study findings.

Author Response

It would seem that reviewer #1 misapprehended the purpose of our study. They incorrectly state that “This manuscript presents a systematic review of the nutritional interventions in cognitive health”, and request alterations that would be appropriate for such a review, such as registration with PROSPERO. Further, they request the exact search string when it has already been provided (page 2, line 79-80 of the materials and methods section). Following their recommendations would result in an entirely different paper. Given reviewer #2 correctly recognised that “This manuscript presents a methodology for identifying and clustering literature in a thematic area.”, we are unsure which element of the manuscript lead to reviewer #1’s misinterpretation. Accordingly, we are unable to implement reviewer #1's suggestions and seek editorial guidance on how to proceed.

Reviewer 2 Report

See attached

Author Response

We have implemented all of the comments provided by reviewer #2 as follows:

 1. In some cases, there is no space between the last letter of a word and the following bracket for a citation.

     We have carefully checked the manuscript and added spaces between words and citations throughout.

 2. Line 10 – replace “educative” with “educational”

    The replacement has been made.

 3. Line 65 – “literature space” is jargon, please use plain language.

    We have simplified the term to ‘scholarly literature’. We have also altered later use of this term in the manuscript (line 246) to avoid the jargon.

 4. Line 80-81 – this sentence (beginning “This yielded….”) is missing a verb or a few

words.

    We have split this sentence into two: the revised sentences can be found on line 82, “This yielded 6,138 citations. This was reduced to 6,045 once the search was restricted to peer-reviewed journal texts.”

5. Line 90-91 – the sentence beginning “When unavailable….” is hard to follow. How

was the existence of a paper known if its title and/or abstract was unavailable from

the search?

     Web of science searches provide the DOI for all entries, in some cases the title field was missing from raw due to disruption from non-askii characters (e.g. atypical semicolons), but much more commonly, the abstract was omitted. Thus papers were identifiable, but portions of their contents needed for later analysis were missing. To help clarify this within the paper, we have added the additional explanatory lead-in, “All titles and abstracts were extracted from Web Of Science search results, which provide digital object identifiers (doi) and additional details.

6. Line 163 – “wait” should be “weight”

    Yes, it should be, and it has been updated as such.

7. Line 185 – insert comma after “clusters”

    Comma has been added.

8. Line 206 – Specify what the “This” starting the sentence is referring to since a new

paragraph has been started.

    We have added additional content to this line as requested, it now reads “This clustering on the basis of age, rather than conceptual content, is not a shortcoming of our analysis, but rather an insight into the genuine structure of the nutrition interventions as they currently sit within the literature.

9. Line 210-211 – awkward wording of this sentence, please revise.

    We have updated this sentence to “A single citation can sit within multiple clusters, therefore it is unlikely that the cluster structure we found is because single interventions include multiple targets”

10. Line 214 – delete “as”

    We have deleted as indicated.

11. Line 215 – change “forming” to “formed”

    We have updated as indicated.

12. Line 221 – use of “Relatedly” is awkward

     We have removed this word.

13. Line 233 – change “This” to “The”

     We have updated this word as indicated.

14. Line 242 – delete “have”, i.e., “…the map we produced….”

     We have deleted this word.

15. Line 243 – change “can” to “could”

      We have updated the word.

16. Line 255 – change “my” to “by”

      Apologies, but we cannot find the ‘my’ requested to be updated.

Round  2

Reviewer 1 Report

My comments to this manuscript are the same as before the revision. The problems remain the same.